# The spread of the first introns in proto-eukaryotic paralogs

Julian Vosseberg [1], Michelle Schinkel [1,2], Sjoerd Gremmen [1] & Berend Snel [1✉]

Spliceosomal introns are a unique feature of eukaryotic genes. Previous studies have established that many introns were present in the protein-coding genes of the last eukaryotic common ancestor (LECA). Intron positions shared between genes that duplicated before LECA could in principle provide insight into the emergence of the first introns. In this study we use ancestral intron position reconstructions in two large sets of duplicated families to systematically identify these ancient paralogous intron positions. We found that 20–35% of introns inferred to have been present in LECA were shared between paralogs. These shared introns, which likely preceded ancient duplications, were wide spread across different functions, with the notable exception of nuclear transport. Since we observed a clear signal of pervasive intron loss prior to LECA, it is likely that substantially more introns were shared at the time of duplication than we can detect in LECA. The large extent of shared introns indicates an early origin of introns during eukaryogenesis and suggests an early origin of a nuclear structure, before most of the other complex eukaryotic features were established.

[1] Theoretical Biology and Bioinformatics, Department of Biology, Faculty of Science, Utrecht University, Utrecht, the Netherlands. [2] Present address: Department of Medical Microbiology, Radboud University Medical Center, Radboud Institute for Molecular Life Sciences, Nijmegen, the Netherlands. ✉email: b.snel@uu.nl

Protein-coding genes in eukaryotic genomes are characterised by the presence of introns[1]. Upon transcription, the introns are removed from the pre-mRNA by the spliceosome and the exons are spliced together to form mature mRNA, which is subsequently exported from the nucleus and translated into a functional protein. There are two types of introns; the vast majority of introns is of U2-type[2], which are recognised and spliced out by the major spliceosome. U12-type introns are removed by the minor spliceosome and comprise less than a percent of introns in eukaryotic genomes[2], with a recently discovered exception of 12% in *Physarium polycephalum*[3]; in many species U12-type introns are completely absent[4].

Ancestral reconstructions have revealed that the last eukaryotic common ancestor (LECA) had a genome with a relatively high intron density compared with present-day eukaryotes[5,6] and a complex major spliceosome with approximately 80 proteins[7]. LECA also had U12-type introns and a minor spliceosome[8]. Eukaryotic evolution after LECA predominantly involved the loss of introns, while only certain lineages including plants and animals experienced net intron gain[6].

It has been established that spliceosomal introns originated from prokaryotic self-splicing group II introns during the prokaryote-to-eukaryote transition[9]. These self-splicing introns can proliferate in the host genome but are rarely present within genes in prokaryotes. The most widely assumed scenario is that the self-splicing introns were introduced in the host genome from the protomitochondrion[10,11] but we previously called other sources possible as well[12]. The emergence of intragenic introns underlined the importance of a nucleus—the defining feature of eukaryotes—to separate transcription and translation for splicing to take place completely prior to protein synthesis[11,13]. Furthermore, the origin of nonsense-mediated decay and the elaboration of ubiquitin signalling are proposed to be defence mechanisms against aberrant transcripts and proteins caused by the spread of introns[14].

Eukaryotes are considered more complex than prokaryotes: cells are much larger and contain multiple membrane-bound compartments. Underlying the increase in cellular complexity during the transition to eukaryotes (eukaryogenesis) was an increase in the number of genes caused by gene transfers and gene duplications[15–17]. Mainly genes involved in establishing and regulating a complex cell and relatively few metabolic genes duplicated during eukaryogenesis[17].

Both the numerous gene duplications and the spread of introns through the genome occurred during eukaryogenesis and their interaction could inform the reconstruction of intermediate stages of this still largely unresolved transition. The relation between proto-eukaryotic gene duplications and introns can be researched by identifying positions of introns that are shared between ancient paralogs. An analysis performed on six eukaryotic genomes almost fifteen years ago identified very few shared intron positions that could represent intron insertions predating gene duplication events[18]. However, a study investigating the evolutionary history of a specific protein family, the spliceosomal Lsm and Sm proteins, found introns shared between multiple pre-LECA paralogs[19]. This implies that introns had spread through the genome before the duplications resulting in these paralogs took place. It also suggests that more of these shared intron positions could be detected in other duplicated families.

In this study we utilise the greatly expanded set of eukaryotic genomes currently available to reassess the relation between the emergence of introns and gene duplications during eukaryogenesis. We detected many more shared intron positions than previous estimates. Our findings have implications for the dynamics of intron evolution and the timeline of events during eukaryogenesis.

## Results

**Intron-rich LECA and predominantly loss after**. To investigate the interaction between the spread of introns and gene duplications during eukaryogenesis we used sets of proto-eukaryotic duplications inferred by two independent approaches: the Pfam domain trees from a recent study[17] (Fig. 1a) and the clusters of eukaryotic orthologous groups (KOGs) that have been used before[18] (Fig. 1b). Intron positions were mapped onto protein alignments and ancestral intron reconstructions were performed using maximum likelihood for each KOG and Pfam domain orthogroup (OG). These reconstructions showed intron-rich ancestors of the eukaryotic supergroups (Supplementary Fig. 1). We estimated an intron density in LECA of 10.8 introns per KOG and 1.9 introns per Pfam OG. Similar intron densities of LECA were obtained when using a tree topology with an unresolved root instead of a root between Opimoda and Diphoda[20] (9.9 and 1.7, respectively). Intron loss occurred frequently throughout eukaryotic evolution, with some lineages losing all introns in our set of genes. Intron gain only had a substantial contribution at certain branches, especially dinoflagellates, which has been described before[21]. These findings for the dynamics of introns from LECA to present-day eukaryotes are fully consistent with a previous study that also reconstructed intron-rich ancestors[6] and with the complexity of the spliceosome in LECA and subsequent simplification in most eukaryotic lineages, as inferred before[22].

**Many intron positions in LECA shared between proto-eukaryotic paralogs**. The relatively high number of introns that could be traced back to LECA underlined the potential to find LECA introns that are in the same position in OGs that stem from a proto-eukaryotic duplication, which we refer to as proto-eukaryotic paralogs. For the KOGs, 19.9% of the 19,184 LECA introns considered had an equivalent LECA intron in at least one paralog (Fig. 1c). This is in sharp contrast to the 1.7% of shared introns found in Sverdlov et al.[18], which is probably due to a combination of the low number of six available genomes that were used and the frequent loss of introns. The percentage of shared introns was even higher for the Pfams, with 34.8% of the 7524 LECA introns in paralogous OGs being shared.

Intron positions shared between proto-eukaryotic paralogs could result from the intron being present prior to duplication and subsequently being passed on to both paralogs. It could also result from two parallel intron gains in the same position (Supplementary Fig. 2a). A shared intron position between two homologous genes that were acquired as two separate genes during eukaryogenesis (e.g., cytoplasmic and mitochondrial ribosomal proteins[23]) must have been the result of parallel intron gains. We compiled a set of separately acquired genes for both datasets and obtained percentages of 5.0% and 5.4% shared LECA introns between separate acquisitions for the KOGs and Pfams, respectively. Notwithstanding the influence of incorrect OG assignment inflating the estimated number of LECA introns shared between separate acquisitions (for example, nearly all sequences with introns shared between KOG0806 and KOG0807 correspond to one Pfam OG (NIT2)), this shows that parallel intron gain is a real phenomenon (Supplementary Fig. 2b). However, the introns shared between proto-eukaryotic paralogs were very likely not only the result of parallel gains (Fisher's exact tests, $P = 5.8 \times 10^{-77}$ (KOGs), $P = 3.1 \times 10^{-220}$ (Pfams)). Another potential explanation for shared introns is transfer of introns between paralogs due to gene conversion[24]. However, this scenario cannot account for the frequent presence of multiple LECA introns in the same position between multiple paralogs (Supplementary Note 1). Instead, most of the shared introns probably represent paralogous introns, which hint at a strong

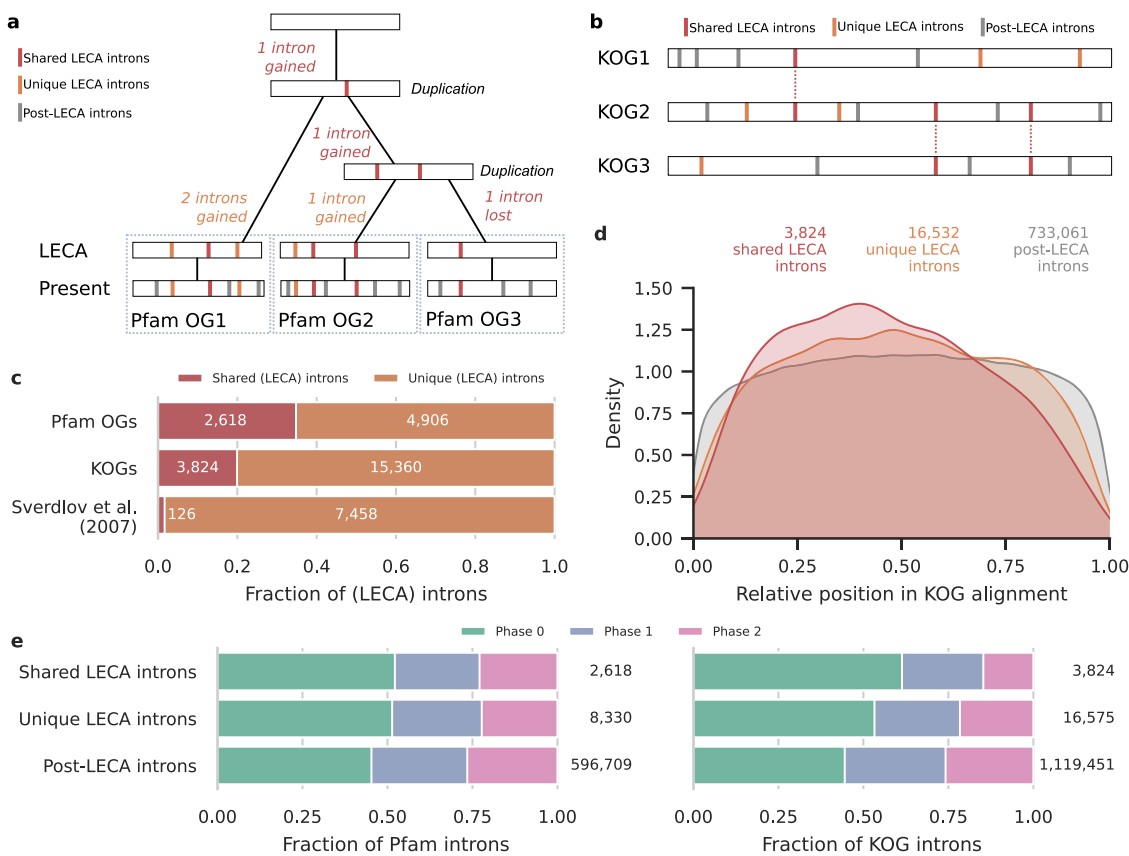

**Fig. 1 Characteristics of unique and shared LECA introns. a** The reconstruction of introns in LECA (Pfam orthogroups (OGs)), distinguishing unique LECA introns and shared LECA introns that likely originated before a duplication. Pfam OG1-3 represent three paralogous Pfam OGs that resulted from two duplications during eukaryogenesis, as indicated. **b** The comparisons of intron positions in KOGs within a cluster, identifying post-LECA introns, unique LECA introns and LECA introns shared between KOGs. KOG1-3 represent three paralogous KOGs in a cluster that resulted from two gene duplications during eukaryogenesis. **c** Fraction of shared LECA introns in the two datasets used in this study, in comparison with the fraction of shared introns as calculated in Sverdlov et al.[18]. **d** Density plot showing the relative positions of introns in the alignment of a KOG. The three distributions are all significantly distinct from one another according to Kolmogorov–Smirnov tests. **e** Intron phase distributions in Pfam OGs and KOGs. All pairwise comparisons were significant, except shared LECA versus unique LECA Pfam introns (Supplementary Tables 1, 2). Numbers in **c–e** indicate the number of introns considered.

association between duplications and intron spread and could elucidate the early spread of introns.

**Intron loss was likely also pervasive before LECA.** To characterise the detected LECA introns shared between proto-eukaryotic paralogs, we compared them with non-shared (which we refer to as unique) LECA introns and post-LECA introns with respect to the relative position of the introns in the gene. Whereas the relative positions of post-LECA introns showed a fairly uniform distribution, unique LECA introns were more at the 5′ end of the gene (compared with post-LECA introns, Kolmogorov–Smirnov (KS) statistic = 0.034, $P_{adj} = 4.1 \times 10^{-16}$) and shared LECA introns were even more biased towards the start of the gene (compared with unique LECA introns, KS statistic = 0.062, $P_{adj} = 5.6 \times 10^{-11}$; Fig. 1d). This bias could reflect preferential intron insertion at the 5′ end specifically during eukaryogenesis or predominant intron loss at the 3′ end. A well-described mechanism of intron loss is by reverse transcription of the intronless mRNA followed by homologous recombination[25]. This mainly affects the 3′ end of the gene, resulting in intron losses from the 3′ to 5′ end. Intron losses before LECA is therefore likely to explain the 5′ bias of LECA introns.

We also compared the phases of the introns, which refers to the three possible positions of an intron in a codon. The phase distribution of the three different categories of introns was also

different (Fig. 1e; $\chi^2 = 966$, df = 4, $P = 8.1 \times 10^{-208}$ (KOGs); $\chi^2 = 182$, df = 4, $P = 2.4 \times 10^{-38}$ (Pfams); Supplementary Tables 1 and 2). LECA introns were more often in phase 0 and less in phase 2 than post-LECA introns. For the shared LECA introns in KOGs this bias was even stronger but in Pfams there was no significant difference between unique and shared LECA introns. The phase distribution differences point to different intron gain or loss dynamics also with respect to phase before and after LECA.

Published phylogenetic trees that were created for the Pfams dataset could in principle help to evaluate the prevalence of intron loss, which in turn might explain the phase and positional biases. We used the topology information in the trees to reconstruct for each duplication node the introns that were likely gained or lost before the duplication. In total, we inferred 999 intron gains and 986 losses before duplications and a further 4906 gains and 1214 losses on the branches that resulted in the LECA families. The phases of gained and lost introns differed slightly ($\chi^2 = 20.0$, df = 2, $P = 4.6 \times 10^{-5}$; Supplementary Fig. 3) but the typical phase bias was inferred for both. This strongly suggests that the phase bias originated from the preferential insertion or fixation of especially introns between codons (i.e., phase 0). For 38% of the duplications, we reconstructed introns being present prior to duplication. For an additional 5% we did not infer the presence of introns in those duplications but we had traced introns in more ancestral duplications. These reconstructions

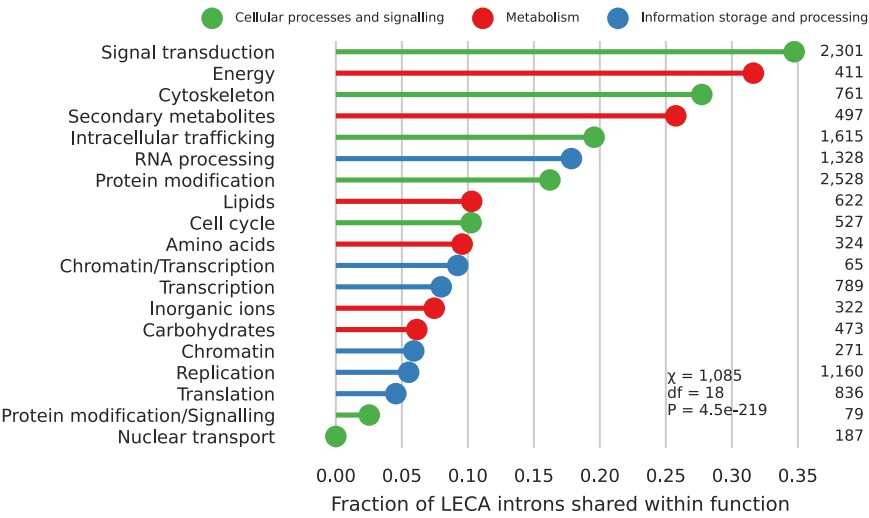

**Fig. 2 Fraction of shared LECA introns between pairs of KOGs in a cluster with the same function.** Sixty-nine percent of pairwise comparisons were significant, including all but one with nuclear transport (Supplementary Data 3). Numbers indicate the number of LECA introns. Only functions with at least ten LECA introns and ten pairs are shown.

strengthen the inference of the dynamic nature of early intron evolution, including the pervasiveness of intron loss already before LECA.

The indications of considerable intron loss prior to LECA suggest that there were initially more pre-duplication introns that were lost and that can no longer be detected. This would mean that the numbers of introns stemming from duplications are underestimates. Moreover, gene families could have been experienced different intron gain and loss dynamics, which means that the absence of detected shared introns should not be seen as evidence that no introns were present prior to duplication.

**Shared introns wide spread across different functions**. Eukaryogenesis was characterised by the complexification of multiple cellular processes and the presence of shared introns between paralogs of a certain function could illuminate how the duplications in that process relate to the spread of introns. Because information on the phylogenetic relationship between the KOGs was not available, we compared the LECA intron positions between KOGs of the same function in a cluster. Intron positions were shared between paralogs of most functions (Fig. 2), mirroring the strong association of duplications and introns. However, appreciable differences between functions could be seen. A relatively large fraction of introns was shared between paralogs in cellular processes and signalling functions, compared with metabolic and informational functions ($\chi^2 = 340$, df = 2, $P = 1.9 \times 10^{-74}$; Supplementary Table 3). The lack of any shared introns out of the 187 LECA introns between the eleven nuclear transport paralogs in the dataset is the most remarkable. The absence of shared nuclear transport introns seems to suggest that a large fraction of these duplications occurred prior to the spread of introns.

For Pfams, the aforementioned reconstruction of intron presence before duplications was used to compare duplications related to different functions. A similar pattern as for the KOGs was observed. Fewer introns preceding a Pfam duplication were inferred for informational and metabolic paralogs than paralogs in cellular processes and signalling functions ($\chi^2 = 46.7$, df = 2, $P = 7.1 \times 10^{-11}$; Fig. 3a; Supplementary Table 4). The large fraction of duplications in energy metabolism with shared introns is almost exclusively due to mitochondrial carrier proteins. Differences in cellular localisation were more subtle ($\chi^2 = 11.0$, df = 4, $P = 0.026$; Fig. 3b; Supplementary Table 5), except for the

high fraction of endosome duplications and the absence of shared introns in cilium duplications. Because nuclear transport is a combination of two functions (nuclear structure and intracellular trafficking), this category is absent in the Pfams set. The ancestral intron reconstructions in the adaptin (Fig. 3c) and SNF2 families (Fig. 3d) illustrate the extent of shared introns, the presence of introns prior to the most ancestral duplication and frequent intron losses in duplicated families. The large extent of shared introns across different functions and localisations implies that introns were present in the genome before much of the complexification of the signalling system, cytoskeleton and endomembrane system and before the full integration of the protomitochondrion into the host.

**Few shared introns between mitochondria-derived paralogs.** During eukaryogenesis, genes were acquired from different prokaryotic sources or arose de novo (i.e., a gene invention). Pfam clades inferred to have been a proto-eukaryotic invention or with an Asgard archaeal or diverse prokaryotic sister group in the phylogenetic tree had the highest fraction of duplications with introns ($\chi^2 = 53.6$, df = 5, $P = 2.5 \times 10^{-10}$; Fig. 4a, Supplementary Table 6). Duplications in Pfam domains that had an alphaproteobacterial sister group were noticeably less likely to have reconstructed introns. When looking only at the first duplication in an acquisition or invention (i.e., the most ancestral one) a similar though not significant pattern was observed ($\chi^2 = 9.2$, df = 5, $P = 0.10$; Fig. 4b). A substantial fraction of the Pfams that were very likely inherited from the Asgard archaea-related host (0.21, [95% CI using the Wilson score interval: 0.13–0.32]) had introns traced back prior to the most ancestral duplication.

The lack of shared introns in certain gene families could be because these duplications had occurred at an early stage during eukaryogenesis when there were no or few introns or could be due to factors such as extensive domain accretion and loss, a low number of LECA introns and intron loss (Supplementary Fig. 4). Due to the inferred pervasive intron loss, it is also more likely to detect shared introns in case of multiple paralogs. By comparing differences between functions and phylogenetic origin in the fraction of duplications with shared introns, the fraction of introns that are shared and the number of introns per Pfam OG, the contribution of these factors can be elucidated (Supplementary Fig. 5, Supplementary Tables 7–9, Supplementary Data 6 and 7). For example, fewer LECA introns were present in

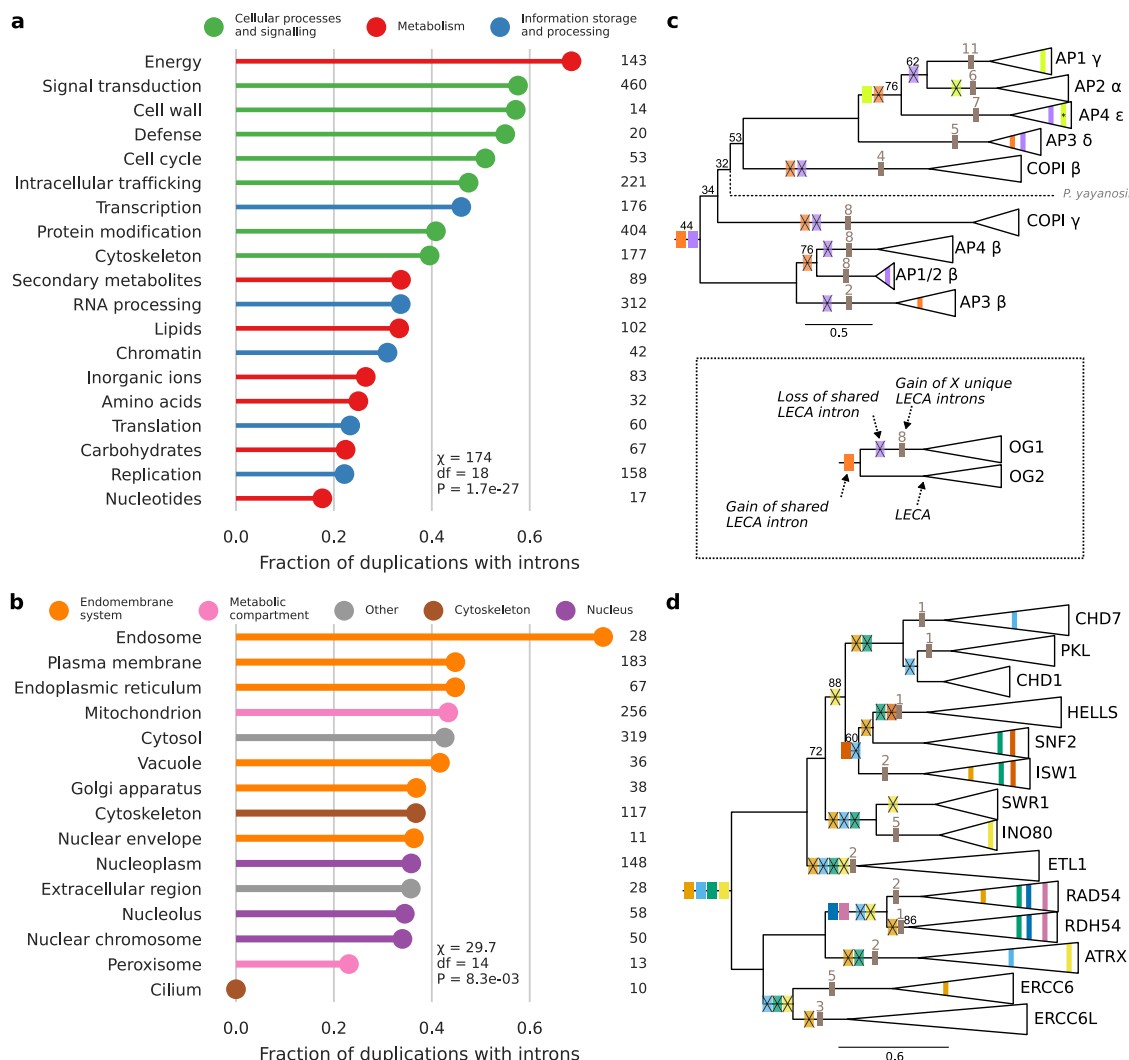

**Fig. 3 Reconstruction of pre-duplication introns in Pfam duplications of different functions and cellular localisations. a**, **b** Fraction of duplications with introns traced to their pre-duplication state according to functional category (**a**) and cellular localisation (**b**). Thirty percent of pairwise comparisons of functions were significant (Supplementary Data 4). 14% of pairwise comparisons of localisations were significant, which were only comparisons including the endosome and cilium (Supplementary Data 5). Numbers indicate the number of duplications. Only functions and localisations with at least ten duplications are shown. **c**, **d** Excerpts from the gene trees of the adaptin (PF01602) (**c**) and SNF2 family (PF00176) (**d**) with the reconstructed presence of introns depicted. The triangles and names correspond to the Pfam OGs. The shared LECA introns in a Pfam OG are coloured and the gain and loss of these introns is mapped onto the phylogeny. The number of unique LECA introns is indicated in grey. Ultrafast bootstrap support values lower than 100 are shown. The branch with a prokaryotic sequence that fell between the Pfam OGs in (**c**) is shown as a dotted line. The shared intron in AP4 ε that is marked with an asterisk was classified as a U12-type intron. Although the phylogenetic position of the two COPI subunits is probably incorrect, the inferred intron gains and losses in these trees are largely unaffected by topology changes.

alphaproteobacteria-related OGs, which would make the detection of shared introns less likely and could explain the low fraction of alphaproteobacteria-related duplications with introns. Despite the relatively large influence of a few clades on differences between groups (Supplementary Note 2), an appreciable number of shared introns was detected for most functions, localisations and phylogenetic origins.

In a previous study, we estimated the timing of duplication events with branch lengths[17]. Based on this branch length analysis, duplications without shared introns were in general older than those with shared introns (KS statistic = 0.064, $P = 0.0016$; Supplementary Fig. 6a). However, the distributions overlap to a very large extent and a considerable number of duplications with introns were relatively old. Almost one-fourth of duplications with introns were estimated to be older than the mitochondrial acquisition. Notwithstanding the uncertainties and

limitations of these branch lengths analyses[26], introns seemed to have been present in the proto-eukaryotic genome from an early stage in eukaryogenesis.

## Discussion

In this study, we investigated the intersection of the emergence of introns and gene duplications during eukaryogenesis. We detected a 12-fold higher fraction of shared intron positions between proto-eukaryotic paralogs in the KOGs dataset than Sverdlov et al.[18] and an even higher fraction in a second independent dataset. The numbers of shared introns were no longer in the range of what is expected from parallel intron insertions, which means that the vast majority of these shared introns were very likely obtained before the duplication. Because our observations hint at a pattern of pervasive intron loss during eukaryogenesis,

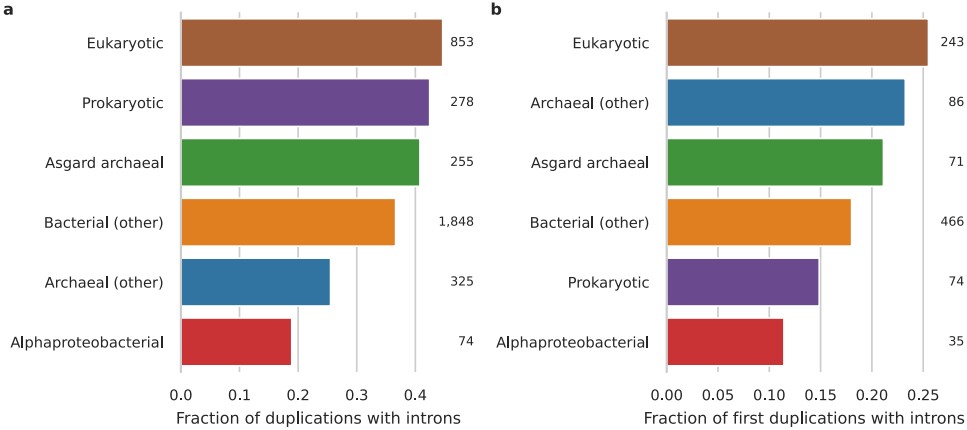

**Fig. 4 Reconstruction of pre-duplication introns in Pfam duplications of different phylogenetic origins. a** Fraction of duplications with introns traced to their pre-duplication state for different phylogenetic origins. Sixty percent of pairwise comparisons were significant, including all but one comparison with alphaproteobacterial duplications (Supplementary Table 6). **b** Fraction of the most ancestral duplication in an acquisition or invention with introns traced to their pre-duplication state. Differences between groups were not significant. Numbers indicate the number of duplications.

the number of introns stemming from duplications that we present are still underestimates.

The higher fraction of shared introns in the Pfams dataset could result from KOGs being undersplit compared with Pfam OGs, which means that multiple bona fide OGs are combined into one OG. This is illustrated by the SNF2 and adaptin examples, for which several Pfam OGs correspond to a single KOG (e.g., AP1G and AP4E in KOG1062, and ERCC6 and ERCC6L in KOG0387). An additional explanation could be that Pfam domains are more conserved. Consequently, it would be more likely for an intron to be paired with an intron in a paralog. Notwithstanding the subtle differences between the two datasets, both revealed consistent findings.

Introns in nearly all species have a phase bias, with most introns in phase 0 and fewest in phase 2. This bias was also present among LECA introns and the bias tended to be even stronger in shared LECA introns. The bias could be due to the preferential insertion or fixation of introns in a certain phase resulting from the overrepresentation of protosplice sites[27] in a certain phase. A biased loss could also explain the phase bias[28]. Another plausible explanation that has been put forward is that the initial distribution was uniform and that the eventual phase bias was due to a combination of massive U12-type intron loss and the directed conversion of phase 0 U12-type to U2-type introns[2]. The phase bias of recent U2-type intron gains in the dinoflagellate lineage and recent U12-type intron gains in *Physarum* lends support to the protosplice site model in at least these eukaryotic lineages[3,21]. The similar phase bias of pre-LECA gains and losses (Supplementary Fig. 3) points to differences already during intron gain, which are then reflected in phase differences during intron loss. Furthermore, the relatively low number of inferred U12-type intron losses and conversions (Supplementary Note 3; Supplementary Figs. 7 and 8, Supplementary Tables 10–12, Supplementary Data 8 and 9) tends to refute a major role of U12-type introns in the phase distribution of all introns. Both observations provide support for the protosplice site model during eukaryogenesis as well.

Our data on the strong association between proto-eukaryotic duplications and introns as compiled here have several implications for the order of events during eukaryogenesis. The large extent of shared introns between ancient paralogs across different functions, subcellular localisations and phylogenetic origins as well as the branch lengths provide consistent evidence for an early origin of intragenic introns during eukaryogenesis, before most of the complex eukaryotic features emerged. In fact, it seems

unambiguous that numerous gene families expanded primarily after the spread of introns through the proto-eukaryotic genome (e.g., SNF2 and adaptin). Other families seem conspicuously devoid of shared introns (e.g., those involved in nuclear transport and the cilium). The conservation of introns upon duplication challenges the main role of retrotransposition in creating proto-eukaryotic paralogs, which results in intronless paralogs and was proposed based on the initial lack of detected shared introns[18]. An early origin of introns should have entailed an early origin of a structure to separate transcription and translation, preventing the erroneous translation of introns into protein. The recent observation of spatial separation between DNA and ribosomes in Asgard archaeal cells tentatively suggests that a separating mechanism may have already been present before eukaryogenesis[29]. The lack of introns shared between nuclear transport paralogs seems to indicate that the emergence of a nucleus with an elaborate nuclear transport system occurred before the wide spread of introns.

A notable exception to the described pattern of shared introns between most categories is the low number of duplications with shared introns in alphaproteobacterial acquisitions, which were very likely present in the protomitochondrion. It is tempting to speculate that these duplications were due to another mechanism; for example, they may have been the result of serial endosymbiotic gene transfers[16]. The protomitochondrion has been widely considered to be the source of introns, even though direct phylogenetic evidence is lacking[12]. Based on the analysis of shared introns, the close integration of the endosymbiont within the host by means of mitochondrial transport seemed to have occurred after substantial spread of introns. Although the symbiosis must have started before the close integration, this observation combined with the inferred timing from our branch lengths analysis is not easy to reconcile with the hypothesis that spliceosomal introns originated from mitochondrial self-splicing group II introns. The self-splicing introns could have come from another lineage instead.

Our analysis was to the best of our knowledge the second large-scale investigation on the association between introns and proto-eukaryotic duplications, yet it was the first to encounter a large-scale occurrence of introns shared between proto-eukaryotic paralogs. Besides the potential implications on the order and causality of events during eukaryogenesis, the strong association between proto-eukaryotic duplications and introns also sheds unique light on the origin and evolution of intron phases and positional biases as well as the discussion on the

emergence of U2- and U12-type introns (Supplementary Note 3). Thus, going forward we expect that further utilisation and understanding of these intertwined processes could be of great help to understand the evolutionary history of individual gene families as well as eukaryogenesis.

## Methods

**Data**. To reconstruct ancestral intron positions we used a diverse set of 167 eukaryotic (predicted) proteomes, as compiled for a previous study[30]. In that study, these proteins had been assigned to the different eukaryotic eggNOG families (euNOGs)[31] using hidden Markov model profile searches[30]. Sverdlov et al.[18] used the homologous clusters of eukaryotic orthologous groups (KOGs) and candidate orthologous groups (TWOGs) from Makarova et al.[15]. KOGs are included in the euNOGs and we used the euNOG corresponding to a TWOG, if present, as determined in Vosseberg et al.[17]. Both types of euNOGs are referred to as "KOG" in the main text. We detected a few differences between the Makarova et al. and Sverdlov et al. clusters and chose one clustering over the other on a case-by-case basis after manual inspection (Supplementary Data 1). The sequences corresponding to these clusters of KOGs were selected and combined per KOG.

We also used the Pfam LECA families and duplications that we published recently[32]. In short, we selected eukaryotic sequences based on best bidirectional hits between Opimoda and Diphoda for tree inference and supplemented these with prokaryotic sequences. In the resulting phylogenetic trees, acquisition, duplication and LECA nodes were inferred. The tree sequences belonging to a LECA node were complemented with the eukaryotic sequences that had one of these tree sequences as their best BLAST[33] hit, resulting in an OG. Sequences from species that are not in the set of 167 species and human sequences that are not in the primary assembly were removed from the OG. If there was only one OG for a Pfam, it was not included.

For predicting the type of introns, genome sequence files of the species in our set were obtained using the links in Supplementary Table 1 of Deutekom et al.[34], with the exception of *Homo sapiens*, whose genome was replaced with the corresponding primary assembly (ftp://ftp.ensembl.org/pub/release-87/fasta/homo_sapiens/dna/Homo_sapiens.GRCh38.dna.primary_assembly.fa.gz), and *Stentor coeruleus*, for which we used the file from NCBI (ftp://ftp.ncbi.nlm.nih.gov/genomes/all/GCA/001/970/955/GCA_001970955.1_ASM197095v1/GCA_001970955.1_ASM197095v1_genomic.fna.gz) to be able to match the sequence file with the genome features file.

**Multiple sequence alignments**. All multiple sequence alignments were performed with MAFFT v7.310[35]. Each KOG was aligned separately using the E-INS-i algorithm and the resulting KOG alignments for a cluster of KOGs were merged into a single alignment (merge option with E-INS-i). If this was not feasible due to memory issues, the alignment was made with the FFT-NS-i option, or FFT-NS-2 if that was also not feasible. Each Pfam OG was aligned separately, followed by a merged alignment of all OGs per Pfam, both with the L-INS-i algorithm. For PF00001 and PF00069, alignments had to be performed with the FFT-NS-i option.

**Mapping intron positions onto the alignments**. We downloaded the genome annotation files from 156 species from our set that we could extract intron information from (Supplementary Data 2). The location of introns was mapped onto the protein alignments using a custom Python script. For each intron position detected in the alignment of an OG, taking into account the three different possible phases, it was determined if at least one sequence of each species had an intron at that position. An intron table was created with per species a string of intron presences ("1") and absences ("0") and a mapping to the position in the alignment of an OG. If an ortholog was missing or intron mapping was not successful, question marks were inserted.

To calculate the relative position of an intron in a gene, sites with 90% or more gaps in the alignment of a KOG were masked. These gap scores were calculated with trimAl v1.4.rev15[36].

**Intron gain and loss rates across the eukaryotic tree of life**. For each branch in the species phylogeny, maximum-likelihood estimates of intron gain and loss rates were obtained using Malin[37] with default settings. The used species phylogeny can be found in Supplementary Fig. 1 of Deutekom et al.[30]. Because the position of the eukaryotic root remains under debate[38], we also used a tree with an unresolved root between Diaphoretickes, Amorphea, Metamonada and Discoba.

**Ancestral intron reconstructions**. The number of introns per ancestral node including missing sites were estimated in Malin and per intron position the probability of the intron being present at a node and gained or lost on the branch leading to a node was inferred. The distribution of posterior intron presence probabilities at the LECA node showed a clear divide between most introns with a very low and a small fraction with a very high LECA probability (Supplementary Fig. 1c, d). For choosing an appropriate threshold to consider an intron a LECA intron, we tried to minimise the effect of misalignment of residues and incorrect

OG assignment on the one hand and to be not too strict on the other hand. This was because an intron with a lower LECA probability that is shared with a paralog, makes it more likely that the intron was in fact present in LECA. Therefore, intron positions with a probability of at least 0.5 were considered LECA introns.

The different KOGs in the KOG clusters do not all represent gene duplications during eukaryogenesis; some were acquired as separate genes[15]. This could be due to separate acquisition events (pseudoparalogs) or the acquisition of already duplicated genes. Shared intron positions in these had to be the result of parallel intron insertions. To identify these, we used the phylogenetic trees of these clusters that we inferred before[32]. If the sequences corresponding to different KOGs were in separate acquisitions and none of the acquisitions in the tree had another acquisition in the inferred sister group, the intron positions in these separately acquired KOGs were compared and shared introns were identified. Introns that were only shared between separate acquisitions were not included in the shared introns analysis. All introns of KOGs that were acquired separately from all other KOGs in the cluster were not used for calculating the fraction of shared introns.

For the Pfams, separate acquisitions were identified based on phylogenetic trees as well using the same approach. For each duplication node in the trees, the intron positions that were present before the duplication event were inferred from the LECA introns using a Dollo parsimony approach. The inferred sister groups of acquisitions and the functional annotation and duplication length information were extracted from Vosseberg et al.[32].

**U12-type intron predictions**. Spliceosomal snRNA genes were searched for in the genomes using Infernal v1.1.2[39] (command used: cmscan -nohmmonly -rfam -cut_ga) with the spliceosomal snRNA Rfam 14.2[40] covariance models RF00003, RF00004, RF00007, RF00015, RF00020, RF00026, RF00488, RF00548, RF00618, RF00619, RF02491, RF02492, RF02493 and RF02494. Introns from species for which none of the snRNA genes specific for the minor spliceosome (U11, U12, U4atac or U6atac) were detected in the genome were annotated as U2. Intron types from the remaining 70 species were predicted with intronIC v1.0.11 + 2.gf7ac7be[2], using all isoforms if needed. Intron positions that were predicted as U12 in at least three species were annotated as U12-type introns.

**Statistics and reproducibility**. Associations between two categorical variables were tested with $\chi^2$ contingency table tests or Fisher's exact tests (in case of $2 \times 2$ tables). When testing the overrepresentation of functional categories, KOGs with multiple categories spanning the three main groups (information storage and processing, cellular processes and signalling, metabolism) were excluded for comparisons between these groups. The numbers of unique LECA introns and shared LECA introns or duplications with and without introns traced back to the pre-duplication state were compared between different functions, cellular localisations and phylogenetic origins. Differences in relative position and branch lengths were assessed with Kolmogorov–Smirnov tests. All performed tests were two-sided and P values from multiple comparisons were adjusted for the false discovery rate. Statistical analyses were performed in Python using NumPy v1.21.4[41], pandas v1.3.1[42], SciPy v1.7.0[43] and statsmodels v0.11.2[44]. Figures were created with Matplotlib v3.4.2[45], seaborn v0.11.1[46], ETE v3.1.1[47] and Jalview v2.11.1.4[48].

**Reporting summary**. Further information on research design is available in the Nature Research Reporting Summary linked to this article.

## Data availability

The data underlying this article are available in figshare, at https://doi.org/10.6084/m9.figshare.16601744[49]. The accession information for the public datasets used in this study is presented in Supplementary Data 2. The source data behind the graphs in the paper are provided as Supplementary Data 10.

## Code availability

The code used to map the intron positions onto the alignments and create the intron tables is available on Github (https://github.com/JulianVosseberg/imapper) and figshare, at https://doi.org/10.6084/m9.figshare.19411820.v1[50].

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

## Acknowledgements
We thank Eva Deutekom for providing the eggNOG annotations and the members of the Theoretical Biology & Bioinformatics group for useful discussions. This work is part of the research programme VICI with project number 016.160.638, which is financed by the Netherlands Organisation for Scientific Research (NWO).

## Author contributions
J.V. and B.S. conceived the study. J.V. and M.S. performed the research. J.V. and B.S. analysed and interpreted the results. J.V., S.G. and M.S. developed the intron mapping pipeline. J.V. wrote the manuscript, which was edited and approved by all authors.

## Competing interests
The authors declare no competing interests.
