## [Peer Review File · Communications Biology]

Reviewers' comments:

Reviewer #1 (Remarks to the Author):

Vosseberg et al., "The spread of the first introns in proto-eukaryotic paralogs"

In this paper the authors describe an analysis of early intron evolution. Their strategy is based on the very elegant idea of comparing the introns of paralogous genes that duplicated prior to the last eukaryotic common ancestor (LECA), with the broad assumption that shared intron positions indicate introns that predate the duplication event (a bit more on that later). They find a large number of LECA introns, more than previous studies, suggesting a very early origin for introns.

Introns were categorised into shared LECA introns, unique LECA introns, and recent/post-LECA introns and they were analysed by location in the gene. They also analysed the LECA introns according to the function of the genes they are within.

Overall I found this to be an interesting paper.

I have a few important points for consideration.

1) With regard to the central assumption that shared introns between paralogs predate the duplication event, the authors consider the possibility of independent intron origins, which they estimate, but they do not consider an alternative, more probable scenario which is gene conversion between paralogy. This is known to be common, and could potentially result in an intron being added horizontally to a paralog. At the very least this must be mentioned as a possibility and the potential impact on the analysis here should be discussed. Ideally, the authors should go further and add some rules to their probabilistic model which would increase the chances that the intron genuinely predates the duplication event.

In my view this is an important point that must be addressed.

2) I am not very convinced by the interpretation of the analysis of the functional categories of genes containing ancient introns and I do not see the rationale for this analysis. The authors should lay this out more clearly. In my opinion, it is quite speculative to assume that the modern day functions, in terms of cellular compartments and processes etc were necessarily conserved back at LECA. A clear example of this is the Pax6/eyeless gene which is an ancient gene that predates the origin of the eye (but is presumed to be indicative of a rudimentary light-sensitive patch in the common ancestor).

This section needs better justification and explanation, and the speculation needs to be clearly separated from the 'results'.

With regards to the functional enrichment analysis, the details in the methods are quite sparse and it is not clear what comparisons are made. I think it is important that the 'LECA introns' genes should be compared to other identified LECA paralogs, and not simply to all other genes, as there may be a discoverability bias for the LECA paralogs, with slower evolving paralogy presumably easier to identify. As rates of evolution have gene function biases, with some processes being more conserved than others, comparing with all genes could create artefactual enrichment results.

Reviewer #2 (Remarks to the Author):

This manuscript describes an analysis of shared introns within pre-LECA paralogs. A similar study was published a decade ago, when too few eukaryotic genomes were available. Since pre-LECA duplications are such an important window into eukaryogenesis, this is an important study.

Overall, the study seems well conducted. However, I have a few doubts about the presentation of the results for which I would need clarification.

1) The key results relate to the fraction of LECA introns that are shared between paralogs (Figs. 2A,B, 3 and 4). However, and this may very likely be my fault, I am not sure I understand what those panels actually show. If I understood well, Figs. 2A,B and 4 show the fraction of gene duplicates that have shared introns (at least one shared intron?). Fig. 3 seems to represent the fraction of LECA introns (in paralogs) that are pre-duplicative/shared by paralogs. To me, the key result is that shown in Fig. 3. Nonetheless, this is shown only for KOGs, not for PFAMs. In addition to being inconsistent, showing the results per "gene duplication" does not take other variables into account (number of introns, etc). Please clarify this issue and show the results per category for the fraction of LECA introns shared between paralogs. I.e. simply what fraction of LECA introns in gene duplicates are shared between the duplicates. This is key, since the major conclusion that given "the low number of shared introns in alphaproteobacterial acquisitions [...] close integration of the endosymbiont within the host [...] have occurred after substantial spread of introns" could be incorrect.

2) Related to this first point, it would be informative to show potential hidden correlates to their results. For instance, if I understood correctly, given the large amounts of LECA intron loss, the chances of identifying pre-duplicative (shared) introns will increase with the number of paralogs per group. If e.g. alphaproteobacterial PFAMs have fewer paralogs per compared group, it will have lower fractions of shared introns in Fig. 4. More evidently, the number and density of LECA introns per group needs to be shown and accounted for, since that will also correlate with the fraction of duplicates with shared introns (but not with the fraction of shared introns per LECA intron).

3) Similar statistics should be provided for PFAM vs KOG groups. It seems quite remarkable (yet concerning) that the density of LECA introns is 10.8 per KOG and 1.9 per PFAM OG.

4) Fig 1A and B could be improved. Duplication events should be clearer (i.e. the lines between OGs show either intron gains or duplications?). Then, showing "LECA" and "Present" side by side can be misleading as it looks like an alignment. Please also clarify in the legend that OG1-3 and KOG1-3 are three paralogous genes belonging to the same OG or KOG (if so).

Reviewer #3 (Remarks to the Author):

This is an interesting, straightforward, carefully executed analysis of the conservation of intron positions in ancient, pre-LECA duplicated genes. The idea of getting insight into the earliest stages of intron evolution through analysis of those ancient paralogs is not new (their Ref 17 as properly acknowledged), but the available database of eukaryotic genomes is now orders of magnitude larger, and the conclusions differ. Many conserved introns are now detectable in these primordial paralogs, supporting the previous conclusions of a high intron density in LECA and very early intron invasion during eukaryogenesis. On the other hand, the results challenge the previous hypothesis on retrotransposition as the principal mechanism of ancestral gene duplications.

I find that the work is quite well done technically, and I see no reasons to criticize the methods. However, some of the conclusions are, in my opinion, more speculative than the language of the current version seems to imply.

In particular, and quite interestingly, the ancient paralogs of apparent mitochondrial origin contain many fewer shared introns than genes of likely archaeal origin. The authors suggest that this could be due to a distinct route of emergence of these paralogs, namely, repeated capture of the same gene from the mitochondrion as opposed to actual duplication within the host genome. This appears plausible but then, the authors also conclude that mitochondria probably were not the source of the initial intron invasion. Instead, they hypothesize that the introns might originate from a different bacterial symbiont. I cannot quite follow this logic, and in any case, the hypothesis is weak and seems to defy Occam's razor by postulating an extra symbiont. Similarly, the idea that the ancestor of the nucleus was already present in the Asgard archaeon that was

ancestral to eukaryotes is tenuous at best. The specific protein markers of the nuclear membrane and nuclear pore so far have not been identified in Asgard genomes despite intense search that yielded many other eukaryote signature proteins (eg. <https://pubmed.ncbi.nlm.nih.gov/33911286/>). In my opinion, all these hypotheses should be tempered. The inferences from intron conservation in ancient paralogs are quite interesting but hardly can be definitive when it comes to staging the early events of eukaryogenesis.

A minor point:

with regard to the origin of the nucleus, in addition to the current Ref. 11, it is necessary to cite <https://pubmed.ncbi.nlm.nih.gov/16615090/>

Reviewers' comments:

Reviewer #1 (Remarks to the Author):

Vosseberg et al., "The spread of the first introns in proto-eukaryotic paralogs"

In this paper the authors describe an analysis of early intron evolution. Their strategy is based on the very elegant idea of comparing the introns of paralogous genes that duplicated prior to the last eukaryotic common ancestor (LECA), with the broad assumption that shared intron positions indicate introns that predate the duplication event (a bit more on that later). They find a large number of LECA introns, more than previous studies, suggesting a very early origin for introns.

Introns were categorised into shared LECA introns, unique LECA introns, and recent/post-LECA introns and they were analysed by location in the gene. They also analysed the LECA introns according to the function of the genes they are within.

Overall I found this to be an interesting paper.

I have a few important points for consideration.

1) With regard to the central assumption that shared introns between paralogs predate the duplication event, the authors consider the possibility of independent intron origins, which they estimate, but they do not consider an alternative, more probable scenario which is gene conversion between paralogy. This is known to be common, and could potentially result in an intron being added horizontally to a paralog. At the very least this must be mentioned as a possibility and the potential impact on the analysis here should be discussed. Ideally, the authors should go further and add some rules to their probabilistic model which would increase the chances that the intron genuinely predates the duplication event.

In my view this is an important point that must be addressed.

The reviewer raises an important concern that is critical to the main interpretation of our results, namely that shared introns between paralogs likely predated the duplication event. In the manuscript we addressed the alternative interpretation of shared introns due to parallel intron insertions by looking for shared introns between genes that were acquired separately during eukaryogenesis. While the reviewer asserts that gene conversion between paralogs is known to be common, we found only three likely cases of intron transfers between paralogs in literature. Nevertheless, this is a very relevant point to discuss. We have therefore carried out additional statistical reasoning on our results in light of intron transfer, acknowledged the possibility of intron transfers in the main text and added a more substantial section about the potential impact on our results in the supplementary text. In short, our statistical reasoning is based on the idea that intron transfer via gene conversion is not expected to affect/include multiple introns or occur multiple times for the same intron. We therefore quantified the number of shared LECA introns in clades that only shared a single intron between two paralogs. The resulting numbers were 148 shared LECA introns in

KOGs (3.9% of shared LECA introns) and 294 in Pfam OGs (11%). A higher number of single shared introns for the Pfam OGs is not unexpected, since Pfam domains are shorter. Because we identified relatively few intron transfer candidates, we argue that most shared LECA introns represent paralogous introns. Looking for example at the adaptin family in Fig. 3, if one wants to explain the presence of shared introns only using introns transfers, four transfer events had to take place, including twice the same introns. For the SNF2 family this would have to be nine single intron transfers, some repeatedly, or six transfers including a block of three introns and a block of two introns. Conversion of such a large part of the gene might as well be called the “duplication” event. For both families, differential loss of ancestral introns is a much more probable scenario in our opinion.

2) I am not very convinced by the interpretation of the analysis of the functional categories of genes containing ancient introns and I do not see the rationale for this analysis. The authors should lay this out more clearly. In my opinion, it is quite speculative to assume that the modern day functions, in terms of cellular compartments and processes etc were necessarily conserved back at LECA. A clear example of this is the Pax6/eyeless gene which is an ancient gene that predates the origin of the eye (but is presumed to be indicative of a rudimentary light-sensitive patch in the common ancestor).

This section needs better justification and explanation, and the speculation needs to be clearly separated from the ‘results’.

We have added the following rationale to the main text to better contextualise why we look at protein function. Eukaryogenesis was characterised by the complexification of multiple cellular processes due to gene duplications. These different processes can be distinguished by considering the functional category. The presence of shared introns between paralogs of a certain function can illuminate how the duplications in that process relate to the spread of introns.

We agree with the reviewer that it is dubious to extrapolate functions in present-day organisms to LECA. For this reason, we only use broad categories, such as the KOG functional categories, which have been used for very similar purposes in previous studies (e.g., refs. 15 and 17; Méheust et al., BMC Biol., 2018; Kauko and Lehto, Proteins, 2017; Pittis and Gabaldón, Nature, 2016). The Pax6 example provided by the reviewer is a transcription factor that is part of KOG0849, whose functional annotation is ‘Transcription’. The homeodomain proteins that comprise this KOG are known transcription factors and this is also the case for nearly all paralogous KOGs in the corresponding homeodomain-like KOG cluster. Given this similar function in all present-day eukaryotes, it is relatively safe to assume that in LECA it also functioned as transcription factor. Similarly, adaptin orthologs in LECA were very likely also involved in vesicular transport, ribosomal orthologs in translation, ubiquitin system orthologs in protein modification, etc.

The implications of our findings according to the paralogous introns interpretation are already partly provided in the results section for readability purposes. The more speculative aspects are separately discussed in the discussion section.

With regards to the functional enrichment analysis, the details in the methods are quite sparse and it is not clear what comparisons are made. I think it is important that the ‘LECA introns’ genes should be compared to other identified LECA paralogs, and not simply to all

other genes, as there may be a discoverability bias for the LECA paralogs, with slower evolving paralogy presumably easier to identify. As rates of evolution have gene function biases, with some processes being more conserved than others, comparing with all genes could create artefactual enrichment results.

The comparisons we performed in the original manuscript follow exactly the principles as laid out by the reviewer, so only between the introns in LECA paralogs. Based on the remark from the reviewer, we provided more details in the Methods section. The Supplementary Tables also show the exact comparisons made.

Reviewer #2 (Remarks to the Author):

This manuscript describes an analysis of shared introns within pre-LECA paralogs. A similar study was published a decade ago, when too few eukaryotic genomes were available. Since pre-LECA duplications are such an important window into eukaryogenesis, this is an important study.

Overall, the study seems well conducted. However, I have a few doubts about the presentation of the results for which I would need clarification.

1) The key results relate to the fraction of LECA introns that are shared between paralogs (Figs. 2A,B, 3 and 4). However, and this may very likely be my fault, I am not sure I understand what those panels actually show. If I understood well, Figs. 2A,B and 4 show the fraction of gene duplicates that have shared introns (at least one shared intron?). Fig. 3 seems to represent the fraction of LECA introns (in paralogs) that are pre-duplicative/shared by paralogs. To me, the key result is that shown in Fig. 3. Nonetheless, this is shown only for KOGs, not for PFAMs. In addition to being inconsistent, showing the results per "gene duplication" does not take other variables into account (number of introns, etc). Please clarify this issue and show the results per category for the fraction of LECA introns shared between paralogs. I.e. simply what fraction of LECA introns in gene duplicates are shared between the duplicates. This is key, since the major conclusion that given "the low number of shared introns in alphaproteobacterial acquisitions [...] close integration of the endosymbiont within the host [...] have occurred after substantial spread of introns" could be incorrect.

The reviewer understood correctly what the panels show. We thank the reviewer for pointing out that it is not easy to understand. To clarify this interpretation of these, we changed the order (of Fig. 2 and 3) and described more explicitly what the differences between the two approaches are. The reviewer raises a valid point regarding the per gene duplication analysis and we included this in the main text. The fraction of duplications with at least one shared intron does not take the number of introns into account, which could mean that for a category with few LECA introns it is less likely to find duplications with shared introns. This cautious interpretation of not finding shared introns was already included in the main text and in the revised manuscript we elaborate more on this regarding shared introns in alphaproteobacterial acquisitions. On the other hand, we disagree that the per intron approach is preferable, because it does not take the duplications and separation into OGs into account, which is crucial for relating duplications to the spread of introns. Both

approaches provide complementary insights into intron dynamics related to duplications during eukaryogenesis. As suggested by the reviewer, the KOG approach was also applied to the Pfam OGs (Supplementary Fig. 5). The results largely agree and a few biologically interesting differences between the per duplication and per intron approach are discussed in the Supplementary text.

2) Related to this first point, it would be informative to show potential hidden correlates to their results. For instance, if I understood correctly, given the large amounts of LECA intron loss, the chances of identifying pre-duplicative (shared) introns will increase with the number of paralogs per group. If e.g. alphaproteobacterial PFAMs have fewer paralogs per compared group, it will have lower fractions of shared introns in Fig. 4. More evidently, the number and density of LECA introns per group needs to be shown and accounted for, since that will also correlate with the fraction of duplicates with shared introns (but not with the fraction of shared introns per LECA intron).

The reviewer raises the point that it is less likely to identify shared introns in case of fewer paralogs given pervasive intron loss. We acknowledge this in the main text of the revised manuscript. The lower number of duplications as well as LECA introns in alphaproteobacterial acquisitions could at least partly explain the lower number of duplications with shared introns that we detected in this category. The average number of LECA introns for different functional groups and phylogenetic origins allowing such contextualisation are provided in Supplementary Fig. 5.

3) Similar statistics should be provided for PFAM vs KOG groups. It seems quite remarkable (yet concerning) that the density of LECA introns is 10.8 per KOG and 1.9 per PFAM OG.

Because KOGs correspond to the entire gene, whereas Pfam OGs reflect a Pfam domain, a lower number of introns for Pfam OGs is expected and not so disconcerting. Furthermore, the observed undersplitting of KOGs, which we mentioned in the main text, results in introns from different bona fide LECA OGs being combined into a single KOG, thereby inflating the intron number. The domain aspect of Pfam OGs is now explicitly mentioned in the main text when introducing the two datasets.

4) Fig 1A and B could be improved. Duplication events should be clearer (i.e. the lines between OGs show either intron gains or duplications?). Then, showing "LECA" and "Present" side by side can be misleading as it looks like an alignment. Please also clarify in the legend that OG1-3 and KOG1-3 are three paralogous genes belonging to the same OG or KOG (if so).

We improved the figure based on the reviewer's suggestions. The splits correspond to duplications and the lines to the paralog branch on which introns could have been gained or lost.

Reviewer #3 (Remarks to the Author):

This is an interesting, straightforward, carefully executed analysis of the conservation of intron positions in ancient, pre-LECA duplicated genes. The idea of getting insight into the earliest stages of intron evolution through analysis of those ancient paralogs is not new (their Ref 17 as properly acknowledged), but the available database of eukaryotic genomes is now orders of magnitude larger, and the conclusions differ. Many conserved introns are now detectable in these primordial paralogs, supporting the previous conclusions of a high intron density in LECA and very early intron invasion during eukaryogenesis. On the other hand, the results challenge the previous hypothesis on retrotransposition as the principal mechanism of ancestral gene duplications.

I find that the work is quite well done technically, and I see no reasons to criticize the methods. However, some of the conclusions are, in my opinion, more speculative than the language of the current version seems to imply.

In particular, and quite interestingly, the ancient paralogs of apparent mitochondrial origin contain many fewer shared introns than genes of likely archaeal origin. The authors suggest that this could be due to a distinct route of emergence of these paralogs, namely, repeated capture of the same gene from the mitochondrion as opposed to actual duplication within the host genome. This appears plausible but then, the authors also conclude that mitochondria probably were not the source of the initial intron invasion. Instead, they hypothesize that the introns might originate from a different bacterial symbiont. I cannot quite follow this logic, and in any case, the hypothesis is weak and seems to defy Occam's razor by postulating an extra symbiont. Similarly, the idea that the ancestor of the nucleus was already present in the Asgard archaeon that was ancestral to eukaryotes is tenuous at best. The specific protein markers of the nuclear membrane and nuclear pore so far have not been identified in Asgard genomes despite intense search that yielded many other eukaryote signature proteins (eg. <https://pubmed.ncbi.nlm.nih.gov/33911286/>). In my opinion, all these hypotheses should be tempered. The inferences from intron conservation in ancient paralogs are quite interesting but hardly can be definitive when it comes to staging the early events of eukaryogenesis.

We thank the reviewer for the kind words about our work. The reviewer raises concerns about two conclusions that are more speculative than we acknowledge, namely about the source of introns and the presence of a nucleus in the host. In both cases we did not intend to draw a conclusion but rather start a discussion. To clarify this intent, we adjusted the phrasing of both hypotheses in the discussion to make their speculative nature clearer and removed the implications on the phylogenetic origin of introns from the final paragraph of the introduction.

We would like to explain why we think these are important discussion points. For the source of the self-splicing introns, we merely wanted to highlight that several of our findings are not readily consistent with the widely believed hypothesis that they came from the mitochondrial endosymbiont. Given the widespread occurrence of horizontal gene transfer in prokaryotes and especially of these group II introns (see <https://doi.org/10.1093/nar/29.5.1238>, for example), it is also plausible that they were acquired by the host lineage before or after mitochondrial endosymbiosis from another prokaryote than the protomitochondrion.

Furthermore, we did not argue that FECA had a nucleus. The remarkable lack of shared introns between nuclear transport paralogs, despite the high number of paralogs and LECA

introns for this function, suggests that a relatively advanced nuclear structure was established before the wide spread of introns through the proto-eukaryotic genome. The exact cell biological nature of the observed separation between DNA and ribosomes in current Asgard archaea is still elusive, let alone if this was also present in FECA. However, because such a separation could make the introduction of an intron into a protein-coding gene less detrimental, it is tempting to speculate that something similar could have taken place in the early proto-eukaryotic lineage.

A minor point:

with regard to the origin of the nucleus, in addition to the current Ref. 11, it is necessary to cite <https://pubmed.ncbi.nlm.nih.gov/16615090/>

We thank the reviewer for pointing out this inadvertent omission. We included the suggested reference.

REVIEWERS' COMMENTS:

Reviewer #1 (Remarks to the Author):

I am happy with the clarifications and responses provided by the authors, and I think the revised ms is good. I have just two small comments.

1. "Moreover, gene families could have been experienced different intron gain and loss dynamics, which means that the absence of detected shared introns should not be seen as evidence that no introns were present prior to duplication."

This statement reminded me of an old paper that I remember by Mikito Go where using GO Plots an ancestral intron was predicted in the precursor to the alpha/beta globins, and eventually an intron was found in the predicted position in plant leghemoglobin. This is an example of a non-shared intron that was inferred to actually be ancestral, so could be cited - I think it strengthens the authors' point. This and other examples are cited and described here: DOI: 10.1016/0378-1119(93)90058-b <https://pubmed.ncbi.nlm.nih.gov/8276250/>

2. P10 "The conservation of introns upon duplication challenges the proposed main role of retrotransposition in creating proto-eukaryotic paralogs18."

I didn't notice this sentence the first time and reading it now I'm not sure what it is supposed to mean. It doesn't seem to be related to the sentences before or after, as far as I can tell. Maybe the point that the authors are trying to make here could be clarified. Trying to interpret what they might mean, I can't see how DNA-based duplication refutes anything with regard to retrotransposition they are just different mechanisms. What is the point that is being argued??

Reviewer #2 (Remarks to the Author):

The authors have addressed my concerns.

Reviewer #1 (Remarks to the Author)

I am happy with the clarifications and responses provided by the authors, and I think the revised ms is good. I have just two small comments.

1. “Moreover, gene families could have been experienced different intron gain and loss dynamics, which means that the absence of detected shared introns should not be seen as evidence that no introns were present prior to duplication.”

This statement reminded me of an old paper that I remember by Mikito Go where using GO Plots an ancestral intron was predicted in the precursor to the alpha/beta globins, and eventually an intron was found in the predicted position in plant leghemoglobin. This is an example of a non-shared intron that was inferred to actually be ancestral, so could be cited - I think it strengthens the authors' point. This and other examples are cited and described here: DOI: 10.1016/0378-1119(93)90058-b <https://pubmed.ncbi.nlm.nih.gov/8276250/>

We looked at the paper mentioned by the reviewer. Because the example does not refer to shared introns between pre-LECA paralogs (the vertebrate alpha/beta globins and plant leghemoglobin trace back to a single LECA globin OG), we decided not to refer to it. Furthermore, the paper uses structural information to predict intron positions in ancestral genes. The correspondence between structural elements of a protein and intron positions has been criticized by others (see for example Stoltzfus et al., 1994, <https://doi.org/10.1126/science.8023140>). Referring to the suggested paper would require some discussion, which – although very interesting – we think is ultimately beyond the scope of our manuscript.

2. P10 “The conservation of introns upon duplication challenges the proposed main role of retrotransposition in creating proto-eukaryotic paralogs18.”

I didn't notice this sentence the first time and reading it now I'm not sure what it is supposed to mean. It doesn't seem to be related to the sentences before or after, as far as I can tell. Maybe the point that the authors are trying to make here could be clarified. Trying to interpret what they might mean, I can't see how DNA-based duplication refutes anything with regard to retrotransposition they are just different mechanisms. What is the point that is being argued??

We thank the reviewer for pointing out the sentence on retrotransposition. The previous paper that investigated the presence of introns that are shared between proto-eukaryotic paralogs found very few. This observation made the authors speculate that retrotransposition was the main mechanism behind gene duplications, because this mechanism creates intronless paralogs. In this manuscript we see the opposite pattern, namely many shared introns. This suggests that other duplication mechanisms than retrotransposition resulted in the numerous paralogs that emerged during eukaryogenesis. We supplemented the sentence in the discussion with a bit more context for clarification.

Reviewer #2 (Remarks to the Author)

The authors have addressed my concerns.